# Charting an Alternative Course for Mental Health-Related Anti-Stigma Social and Behaviour Change Programmes

**DOI:** 10.3390/ijerph191710618

**Published:** 2022-08-25

**Authors:** Daniel Walsh, Juliet Foster

**Affiliations:** Institute of Psychiatry, Psychology & Neuroscience, King’s College London, London SE5 8AB, UK

**Keywords:** public health, stigma, mental health, behaviour change, culture, communication, mixed-methods, data science

## Abstract

Mental health-related anti-stigma strategies are premised on the assumption that stigma is sustained by the public’s deficiencies in abstract professional knowledge. In this paper, we critically assess this proposition and suggest new directions for research. Our analysis draws on three data sets: news reports (N = 529); focus groups (N = 20); interviews (N = 19). In each social context, we explored representations of mental health and illness in relation to students’ shared living arrangements, a key group indicated for mental health-related anti-stigma efforts. We analysed the data using term-frequency inverse-document frequency (TF-IDF) models. Possible meanings indicated by TF-IDF modelling were interpreted using deep qualitative readings of verbatim quotations, as is standard in corpus-based research approaches to health and illness. These results evidence the flawed basis of dominant mental health-related anti-stigma campaigns. In contrast to deficiency models, we found that the public made sense of mental health and illness using dynamic and static epistemologies and often referenced professionalised understandings. Furthermore, rather than holding knowledge in the abstract, we also found public understanding to be functional to the social context. In addition, rather than being agnostic about mental health-related knowledge, we found public understandings are motivated by group-based identity-related concerns. We will argue that we need to develop alternative anti-stigma strategies rooted in the public’s multiple contextualised sense-making strategies and highlight the potential of engaging with ecological approaches to stigma.

## 1. Introduction

The reproduction of mental health-related stigma in society is detrimental to public health [1,2,3,4]. We conceptualise mental health-related stigma as the ‘marks which degrade’ individuals with experiences of mental health problems [5,6,7]. Through the reproduction of stigma in society, individuals with experiences of mental health problems continue to face profound discrimination in multiple domains, including healthcare, education, the media, and the home [1,8,9].

Public health approaches to mental health-related stigma are predominately education-based (e.g., Mental Health Literacy) and sometimes conducted with elements of social contact [1,2,10,11]. Campaigns are premised on two principal assumptions. First, practitioners consider the public stigma to be sustained by the public’s perceived deficit in professional knowledge [2,12]. Second, public knowledge is held to have an abstract quality in opposition to the common-sense understandings developed in the rhythm of everyday living [2,12,13]. Time to Change is emblematic of this way of tackling mental health-related stigma. It was Britain’s predominant social marketing campaign between 2007 and 2021 [4]. In its last phase, it combined humorous videos of male friendship (i.e., parasocial contact) with frequency statistics (i.e., mental health literacy) to promote inter-group empathy [4].

Overall, reviews suggest that anti-stigma programmes increase the public’s mental health-related knowledge and positive attitudes towards professional services in the short-term, especially when programmes are targeted to key groups, educate in non-categorical models of mental illness, include elements of social contact, and are culturally relevant [2,10,11,14,15,16]. However, despite hopes that short-term attitudinal changes might be attenuated, there is limited evidence for sustained social and behavioural change, and there are serious concerns about campaigns’ unintended consequences [2,8,10,14,17]. Concerns regarding the unintended effects of solely education-based interventions are well established, especially those which focus on biogenetic explanations of mental health and illness [2,8,15,18,19,20]. Reviews of national time-trend and experimental studies find an increase in biomedical explanations of mental illness to correlate with an increased desire for social distance in the public [2,4,18].

Social contact is broadly considered the ‘state of the art’ [10] for mental health-related anti-stigma efforts, although disagreements remain (e.g., Jorm, 2020a). Overall, efforts to promote trust through social contact between individuals with and without experiences of mental health problems may be more effective than stand-alone education-based interventions [1,2,3,21,22]. Consistently, nationally representative surveys find that self-reported social contact histories correlate with lower mental health-related stigma measures [4,11,23]. Moreover, expressions of ‘non-familiarity’, ‘incomprehensibility’ and ‘un-knownness’ may constitute a canon in representations of mental illness [24,25,26,27]. However, experimental evidence suggests that negative intergroup contact experiences in a university setting are associated with increases in social distance, disgust and fear [28,29,30,31]. Furthermore, ethnographic evidence suggests that historically-rooted fears of violence and contagion may be sustained even after multiple years of contact in the home [26].

From the outset, there have been disagreements in the literature on social contact [32]. Indeed, a continuance in the history of contact-based interventions is the frustrated search for mediators and moderators that can explain the multiple relationships between social contact and stigma [32]. We lack consensus regarding the content, duration, delivery, frequency and measurement of contact-based interventions, and theorists have questioned whether ‘contact’ can be considered an active ingredient for change [2,32,33]. This gap in understanding limits our ability to translate contact theories into practice, including impact evaluation and improvement [2,34,35].

We need to engage with the complexities of representations of ‘contact’ [2]. Contact-based interventions employ a simplistic model of social cognition: they largely assume that the perception of someone with experiences of mental health problems will overcome a stigmatising perception of mental health problems as ‘unknown’ [2,32]. However, it is questionable how far representations of mental health and illness are unknown to the public. Experiences associated with mental ill-health are common in the public, such as hearing voices, hopelessness and worry, and lay communications commonly reference professionalised terminology and stigma explicitly [36,37,38,39,40]. Furthermore, the western canon is replete with narratives, iconography, and myths related to mental health and illness and its problematic linguistic forebearers of ‘madness’ and ‘insanity’ [25,41,42,43].

Representational ambivalences relating to ‘knowledge’ and ‘contact’ may be rooted in social history. Social histories of taboo highlight how it is common in western history for prohibitions to be placed on intergroup contact, especially on forms of contact associated with in-group identity [44,45,46,47]. Individuals with mental health problems likely share social histories of exclusion common for groups also marginalised in the social order (e.g., Roma, Jews, LGBTQ+) [9,48]. Rather than being literally ‘absent’ from western European history, these groups have suffered contact prohibitions, which may partially remove them from dominant-groups direct perception whilst also remaining hypervisible in stigmatising and often sensationalised communications (e.g., educational materials, local myths, newspaper reports) [24,49,50,51]. Researchers have shown that these contact prohibitions, and the ambivalent perceptions they sustain, enable spatially and temporally remote forces to influence the in-person experience of intergroup contact between individuals perceived to have or not have experiences of mental illness, such that perceptions of mental illness as ‘known’ and ‘unknown’, ‘close’ and ‘distant’, may coexist in lay representation [26,28,49,52,53].

In response to the limited and unintended effects dominant mental health-related anti-stigma efforts have had, in this paper, we will critically engage with the assumptions underpinning dominant mental health-related anti-stigma efforts. To progress beyond the limitations of current social and behavioural strategies, this paper will provide an in-depth analysis of representations of intergroup contact related to mental health and illness. In the remainder of this introduction, we will explain the key theoretical-methodological concepts guiding the construction of procedures and analysis to foreground the methods and results. These are social context, group identity and cognitive polyphasia. For a full explanation of the methodological-theoretical considerations, please see Walsh & Foster (2022).

### 1.1. Social Context 

‘Social context’ refers to the specific setting where social interaction occurs [54]. Public health efforts have a greater impact when they respond to the social contexts through which key groups sustain mental health-related stigma [4,16,55,56]. Whilst there is a broad appreciation for the need to address the social context of stigma, the dominant approach in public health commonly considers content outside of the process of social and behavioural change [57,58,59,60]. Specifically, mental health-related anti-stigma efforts typically reduce social context to individuals’ perceptions of other people’s attitudes and beliefs [2,58]. This is inadequate, as it neglects how axioms of time and space are foundational to the public’s perceptions of mental health and illness [13,53,61]. For example, during the situated experience of intimate contact in a university setting with someone perceived to have a mental health problem, people distanced representations of mental health problems as ‘foreign’, ‘unknown’ and ‘unfamiliar’ in response to concerns for the Self and one’s in-group [28]. Similarly, public concerns about intergroup contact show remarkable historical consistency: common stigmatising images, metaphors and symbols reproduce socially shared representations of mental illness [26,27,58]. 

Theorists have critiqued the assumption that social context can be described as an externality independent of internal psychological functions [50,62]. Indeed, representations of social context are likely motivated by the public’s desire to represent mental illness as ‘not-me’ or ‘Other’ [50]. These motivated representations may be communicated non-consciously through the public’s ritualised effects, cognitions and behaviours [26,28,63,64,65]. Indeed, Othering is thought to be reproduced in the often-latent ambivalences present in communication [58,65,66]. Communication comes in multiple forms and may simultaneously be intra-psychological (e.g., within themselves), interpersonal (e.g., conversations with friends and family) and institutional (e.g., mass media) [64]. At the level of representation, othering may also be reproduced in the meanings that constitute places, such as the home, hospital and university [27,41,43].

### 1.2. Group Identity 

We define group identities by the representations that enable people to identify and be identified [67]. Group identities inform mental health-related anti-stigma campaigns [8,12,68]. Namely, rather than being agnostic about mental health-related anti-stigma campaigns, groups commonly advance motivated and particularised understandings of mental health and illness [2,8,12,68]. For example, we can identify this in the continued authority attributed to mental health professionals in operating education and contact-based mental health-related anti-stigma programmes over lay and service user expertise [2,8]. Furthermore, mental health professionals tend to prioritise educating the public on biogenetic and neurological explanations of mental illness, despite strong objections from service user groups who tend to emphasise the everyday social determinants of stigma [69]. 

### 1.3. Cognitive Polyphasia

Social context and group identity are interdependent [70,71]. As Marková (2008) explains: *“It is not simply that different groups and different social contexts affect what people represent. It is the interactive interdependence between them that produces different styles of thinking and communicating”* (Marková 2008, p. 479). A theory of cognitive polyphasia provides a basis to conceptualize how groups relate to social context [58,72]. Cognitive polyphasia describes how, at the level of representation, individuals and groups engage multiple types of knowledge or beliefs to function within their particular social context [70]. In contrast to theories of cognitive dissonance, theories of cognitive polyphasia emphasise how, in embodied practise, alternative or theoretically contradictory knowledge rarely elicits discomfort within the Self [73]. Researchers have repeatedly found that people make sense of mental health and illness through polyphasias, such as differentiation in explanations of mental illness according to location (e.g., internal vs. external), cause (e.g., biomedical vs. traditional), disorder ‘type’ (e.g., depression vs. schizophrenia) and measurement (e.g., categorical vs. dimensional) [28,38,70,74]. 

Whether contemporary lay mental health-related polyphasias may be rooted in alternative epistemologies (i.e., alternative origins of knowledge) remains unexplored. For the last century, dynamic and static epistemologies have co-existed in the psy-disciplines, although they are practised with varying degrees of authority in different times and spaces [46,71]. In dynamic approaches, social contexts are theorised to motivate fluctuations in people’s movements, cognitions and effects, and pay close attention to individuals’ often idiosyncratic life histories [46,71]. In contrast, in static approaches, human experience is largely generalized or ahistorical and is explained by stable factors [46,71].

### 1.4. Methodology

So far, drawing on a theory of cognitive polyphasia, we have emphasised motivated representations, which relate social context to group identity. We will now expand on how we constructed our procedures. In contrast to the proposition that public understandings of mental health and illness are ‘deficient’ or ‘abstract’, we explored how mental health and illness are represented in relation to the ‘home’ [58]. Cross-culturally and trans-historically, beliefs about contact in perceived intimate spaces are especially resistant to change and commonly elicit the less explicit and socially undesirable meanings present in public understanding [23,26,28].

As we were situated in our representations and the expression of mental health-related meanings is often subtle and ambiguous, we were concerned that we might not fully comprehend these meanings in their complexity [58]. Accordingly, in this paper, we drew on advances in automated natural language processing (NLP) techniques to explore the manifest and latent meanings in public representations of mental health and illness [58,63]. 

Term-frequency inverse-term-frequency (TF-IDF) models are foundational to common NLP techniques [75], as they model which words are used frequently in some but not all documents (e.g., interview transcripts) [76]. This helps discern ‘what’ is being spoken about [58]. However, TF-IDF models are insensitive to the contextually specific, implied, or contradictory meanings present in representation, highlighting the continued need to combine automated approaches with deep-qualitative readings of the text [58,77]. In this sense, TF-IDF models may ‘spotlight’ possible meanings in the text to be interpreted as part of the broader mixed-methodological research practise [58,77]. 

We used the context of students’ shared living arrangements to draw out the public’s sense-making processes. Students are indicated as an important group to target for mental health-related anti-stigma efforts [8,10,56,78,79], as: (1) students tend to be younger and so their representations may be more sensitive to change and intervention effects may be attenuated over the life span; (2) students’ access to education may afford them greater power to challenge societal forms of mental health-related stigma; (3) students have an elevated likelihood of at some point having personal experiences of, or close contact with, psychological distress [8,80,81].

### 1.5. Contributions

In this paper, based on our data, we will make the following arguments:People represent mental health and illness using dynamic and static epistemologies;Polyphasias in representations of intergroup contact are functional to the social context;Expressions of stigma are motivated by group identity.

These findings will challenge the continued utility of dominant mental health-related anti-stigma interventions, and we will suggest alternative directions for research. 

## 2. Materials and Methods

Our analysis draws on three data sets collected between 2019 and 2021: news reports (N = 529); focus groups (N = 20); interviews (N = 19). Each of these data sets was concerned with representations of students’ shared living arrangements concerning mental health and illness and were subjected to the same analytic procedures. We will now explain how we collected and analysed the data. 

### 2.1. Data Collection 

#### 2.1.1. News Reports

We included digital news reports on students’ accommodation concerning mental health and illness. In 2018, 82% of UK 16–24-year-olds reported that they primarily received their news online, either directly from news organisations or indirectly through social media [82]. Therefore, we included the most popular news outlets for this age range: BBC, ITV, Sky News, LADbible, Youtube News, The Guardian/Observer; Channel 4 News, BuzzFeed, the Daily Mail, Google Search, and Google News [82]. To do so, we used the news aggregator LexisNexis and searched directly on the news outlets’ websites. To be included, the news report must contain either ‘student’, ‘university’ or ‘college’ and also ‘housing’, ‘accommodation’, or ‘flat’. Moreover, they must have been written in English and published between 1 January 2010 and 1 January 2020. The following were excluded: duplicates; university websites; blogs; business websites; press releases; sponsored/paid content; specialist professional news outlets (e.g., construction). This produced a corpus of 529 news reports (Table 1). 

#### 2.1.2. Interviews and Focus Groups

Students were recruited to take part in one-to-one interviews and small focus groups. In both, students were invited to explore how people think about shared accommodation, particularly in relation to mental health problems. All the interviews and focus groups were conducted in a private space within the university setting. Students were renumerated £7 per hour for their time.

A convenience sample of 38 students was recruited (Table 2). Posters advertising understanding students’ beliefs about sharing accommodation and mental health” were placed around common student areas and emails were sent to relevant student mailing lists. Potential participants were invited to contact the first author by email. The first author responded to potential participants with an information sheet that detailed the study’s rationale, methodology, potential harms/benefits and recruitment criteria. If the potential participant was interested, a convenient time arranged. Participants were also sent a copy of the consent form. There was at least 24 h between reading the consent form and providing written consent. Participants were free to withdraw at any point. No participants withdrew from the study.

All participants were over 18 years old and were registered undergraduate or postgraduate students at a large urban university in the UK. As previous research has shown that understandings about mental health and illness are differentiated by personal experiences of mental ill-health and cultural background [2,28,83] participation was limited to students with a Home/EU fee status and those who self-reported no current or previous experiences of mental health problems. Approval was obtained from the university research ethics office for the interviews (LRS-18/19-9068) and focus groups (LRS-19/20-14053), including the procedures for ensuring informed consent. 

The 18 of whom took part in the interviews (N = 18; female: 11 (61.1%); male: 7 (38.9%)), and 20 in the focus groups (N = 20; female 17 (85.0%); male 3 (15.0%)). This gender disparity is typical in qualitative studies and focus groups on student mental health-related stigma specifically [84]. While some researchers have opted to use purposeful sampling, others have highlighted that men willing to participate in focus groups may not be representative [84]. As the analysis is primarily concerned with the general ways of making sense of mental health and illness, it is unclear that the gender balance is of immediate concern, although we encourage also exploring gender-specific experiences of mental health-related stigma [37].

Students were arranged into focus groups of three to four students, and six focus groups were conducted in total. About half of the total sample studied psychological or health sciences. No student took part in both the interviews and the focus groups. Previous literature suggested that a sample of this size would be sufficient to achieve theoretical saturation for the study’s qualitative components and produce a corpus large enough for quantitative modelling [77,85,86]. 

The interviews and focus groups were broadly structured using the same topic guide. The topic guide consisted of four broad units: (1) exploring participants’ beliefs, preferences and experiences of student accommodation; (2) exploring participants’ understandings of mental health and illness; (3) exploring participants’ mental health-related forms of self-understanding (4) exploring participants’ beliefs, preferences and experiences of living with someone with a mental health problem. In unit 4, we used Jodelet’s (1991) ethnography as stimulus material to support participants in exploring their beliefs. [26] described a French programme where some patients at a psychiatric hospital lived as lodgers in local family homes. It was common for participants to have heard of this specific programme or ones like it. Reflecting on this programme was useful for helping participants think through some of the subtleties of how they, or those close to them, might feel about living with someone with a mental health problem. The first author conducted and audio-recorded the interviews and focus groups.

We will now provide extra details relating to the focus groups only. Potential participants were purposefully sampled evenly according to whether they would or would not be willing to live with someone with a mental health problem. We decided to group and inform like-minded participants, as we felt this might allow them to express less socially normative beliefs. Two extra sets of stimulus materials were included to support focus-group conversation. At the beginning of unit 2, participants discussed five short publicly available videos (c.a. 30 s). These videos were released in 2017 as part of the third phase of the Time to Change campaign, the preeminent mental health-related national anti-stigma campaign in the UK at the time of data collection [55]. Each of these videos follows the friendship between two young men, states that 1 in 10 young people will experience a mental health problem this year and encourages the public to listen to each other non-judgmentally [55]. At the end of the fourth unit, participants were presented with a reported and intended behaviours scale (RIBS) [87]. RIBS contains questions measuring the public’s prior and projected forms of contact, including whether they would be willing to live with someone with a mental health problem, and is a principal instrument used in the evaluation of the time-to-change campaign [2,87]. After completing the scale, participants shared any thoughts the scale elicited but were not asked to share their specific answers. Subsequently, we adapted the wording used in the RIBS questionnaire so that participants answered questions about prior and projected contact with someone with experiences of depression and schizophrenia specifically and then shared their thoughts with the group. It is standard in public health approaches to mental health-related stigma to compare the public’s perceptions of depression and schizophrenia to develop an account of their unified representations of mental illness [11,23]. We surveyed participants’ behavioural intentions to draw out their mental health-related sense-making processes. However, as previous literature has questioned the validity of using these self-reported measures in predicting the public’s inter-group contact behaviours [28,88,89], they were used solely as stimulus materials and not as a quantitative measure. 

### 2.2. Data Preparation 

Interview and focus-group audiotapes were transcribed verbatim by the first author. The transcription process followed the principle of parsimony to facilitate the reconstruction of participants’ reality in text format [90]. During this process, personal identifiers were removed [90]. A copy of the data was retained in its completeness for subsequent qualitative analysis [90].

We adapted the recommendations set by Bafna and colleagues (2016) to prepare the three corpora—interviews, focus groups and news reports—for quantitative modelling. Bafna and colleagues (2016) have shown their approach to effectively derive TF-IDF scores on multiple forms of unstructured data, including news reports and personal communications and work on small and large data sets. In Python 3, the data was preprocessed. All non-verbal features (e.g., images, emoticons, hyperlinks), text stamps, non-alphabetic characters, punctuation and stop-words were removed, and all the remaining words were made lowercase [85]. 

### 2.3. Data Analysis

Using Python 3, we formulated term-document matrices by applying a TF-IDF algorithm with sub-linear term-frequency scaling to each corpus, and we calculated a cosine-distance matrix [85]. Next, we ranked the TF-IDF scores. Whilst there are no strict cut-off points for using rank scores, it is common practise to present the top 10 [58]. TF-IDF rank score words provided gave a naïve indication of what might be focal in the public’s representation [58]. To explore the meanings involved in representation, we searched the corpora for TF-IDF rank score keywords and produced cross tables matching TF-IDF ranked words with verbatim quotes for each corpus. This is standard in mixed-methodological corpus linguistics approaches to understanding health and illness. Finally, supported by cross-tables, we explored the complexity of meanings present in the data. This was an iterative procedure of developing our account of both what was focal in participants’ imagination (i.e., what they talked about) and what were the social processes involved (i.e., how they talked about it) [63,91]. Close attention was also paid to the latent processes involved in sense-making, including what was implied or absent from the texts and which spaces were drawn upon in representation [26,28,63,65]. 

### 2.4. Triangulation and Validity

We employed triangulation to integrate the results from the three corpora [90]. We explored possible differences and similarities in how mental health and illness were represented in the media, one-to-one interviews and small focus groups [90]. We iteratively progressed this process of triangulation until no new insights into the problem space were further clarified [90].

Validity was established through triangulation [90]. However, we did not directly compare outputs from the quantitative modelling across contexts as graph measures can still produce spurious results when comparing scores calculated on data produced under different social conditions [58,76]. We felt this potential issue was better remediated within the triangulation design [77,90]. Accordingly, quantitative results were not considered in isolation and were interpreted through a close reading of the text [77].

## 3. Results

To review, we applied TF-IDF models to each corpora to ‘spotlight’ possible meanings in the text, and these words were interpreted by resituating them in verbatim quotes. This process of triangulation progressed iteratively, comparing within and between corpora. In this section, we will explore our data and critically engage with assumptions underpinning dominant mental health-related anti-stigma efforts. Interpreting the data using theories of cognitive polyphasia, we will show that lay understandings of mental health and illness are rooted in both static and dynamic epistemologies. In contrast to the assumption that public understandings are held in the abstract, close examination will show these understandings to be functional in the social context. Furthermore, rather than lacking professional knowledge, we will evidence the public to be tenacious in their ability to redevelop multiple understandings of mental health and illness to sustain representations of mental health and illness as Other or ‘not-me’. To support comprehension, we will first present data in the form of TF-IDF rank tables, and then theorise the data in terms of cognitive polyphasia, social context and group identity. We will triangulate throughout.

### 3.1. Epistemology

#### 3.1.1. Summary

Our survey of TF-IDF ranked score words (Table 3, Table 4 and Table 5) suggested that the public represented mental health and illness as a negative social phenomenon. Rather than being considered in the ‘abstract’, people’s representations of mental health and illness were constituted by multiple practical and personal concerns (Table 3, Table 4 and Table 5). We found that people’s understandings of mental health and illness were characterised by polyphasia rather than dissonance: dynamic and static epistemologies commonly co-existed in communication without signs of tension (Table 3, Table 4 and Table 5). Furthermore, we found that common lay polyphasias were grounded in these alternative epistemologies, including ‘social’ and ‘non-social’ explanations of mental illness, as well as comparing perceived disorder and treatment types (Table 3, Table 4 and Table 5).

#### 3.1.2. Dynamic Epistemology

A dynamic epistemology was engrained in the public’s sense-marking about mental health and illness. Expressions of a dynamic epistemology were attendant to people, space and time (Table 3, Table 4 and Table 5). Lay models link these latent axioms in representation. For instance, our close analysis suggested that the remains of a high-profile housing estate in which 72 people died were represented as an ongoing mental health issue that ‘risks’ ‘traumatising’ the ‘wider population’ through its continued public perception (Table 3, rank 8). Furthermore, drawing on a dynamic epistemology, participants emphasised the affective uncertainty of social life and human experience. For example, in one focus group, FG3P3 conceived of “mental health” as “oscillating between very different extremes of feeling” (Table 4, rank 6). People represented these experiential fluctuations as “triggered” by novel person-environment interactions (Table 4, rank 3). For example, in another focus group, FG6P2 emphasised that if there is a “person in your environment” whose capacity to “work” changes, “people could be resentful of this very, very quickly” (Table 4, rank 1). Perceived uncertainty was linked to motivation and action. We can identify this in P3, who associated “you don’t know what they can do” with a desire to protect one’s ‘family’ and communicative taboos (Table 5, rank 3). Alternatively, P5 associated the ‘randomness’ of ‘mood’ with a complete loss of motivation and restriction: “you give up hope in yourself in those moments … and I’m like, what’s the point of doing this” (Table 5, rank 7).

#### 3.1.3. Static Epistemology

People also made sense of mental health and illness using a static form of knowledge (Table 3, Table 4 and Table 5). In this, people largely conceptualised individuals with experiences of mental ill-health as an undifferentiated group to be managed through generic treatment programmes (Table 3, Table 4 and Table 5). This is clearly illustrated in the discussion of how to “protect students’ mental health” (Table 3, rank 3). Here, students were represented as differentiated from the ‘general population’ by their increased likelihood of experiences of “depression” and “loneliness” (Table 3, rank 3). Similarly, the contention that mental health and illness are constituted by generic risk factors is exemplified in the statement: “digital data analytics systems could help universities identify those at risk” (Table 3, rank 9). In the focus groups (Table 4), we can further identify the importance of programmatic thinking in the public’s sense-making about mental health and illness. For example, drawing on NICE guidelines, FG5P1 conceived of mental illness in terms of “lines of treatment”, in which “psychotherapy” is tried before “harder” interventions, such as psychiatric “drugs” (Table 4, rank 5).

#### 3.1.4. Cognitive Polyphasia vs. Cognitive Dissonance

The common practise of contrasting perceived disorder and treatment ‘types’ may be grounded in both dynamic and static epistemologies. Yet, there is limited evidence that these alternative epistemologies were a source of discomfort, as would be expected by theories of cognitive dissonance. For example, the way that FG5P1 contrasted perceived ‘soft’ and ‘hard’ treatments suggests that static models are infused with a dynamic concept of power (Table 4, rank 5). FG6P1 rendered explicit this power inequality (Table 4, rank 7). They contrasted perceived types of disorder to explain FG6P3′s uncertainty about mental ill-health, arguing that for “schizophrenia” rather than “depression”, the “only real solution is drugs” (Table 4, rank 7). Similarly, even though the author’s account of “digital data analytics” is a clear example of a lay static conceptualisation of mental ill-health, in the same sentence, the author also used the word “pressures”, a hallmark of dynamic models, to convey their belief that “mental health concerns” are “rising” (Table 3, rank 9). Indeed, rather than these two alternative epistemologies being experienced in contradiction, the two were subtly intertwined in representation, as predicted by theories of cognitive polyphasia.

### 3.2. Social Context

#### 3.2.1. Summary

The human capacity for cognitive polyphasia enables people to make sense of themselves and others in context. To recap, we explored students’ shared living arrangements in relation to mental health and illness in three social contexts: (1) the wider media, (2) small focus groups and (3) one-to-one interviews. We found that epistemic polyphasia responded to the level of analysis such that concerns for perceived inter-group contact were functional in context. Specifically, through polyphasias in the media, mental health and illness were represented as societal harm, both personally relevant to the public and at the boundaries of public comprehension (Table 3). Alternatively, in focus groups, individuals with experiences of mental illness were represented as an out-group and were degenerated for their perceived deficiencies (Table 4). Our qualitative analysis of the interviews will highlight the theoretical tensions presented by the public’s sense-making processes (Table 5). Whilst they considered themselves different from individuals with experiences of mental illness, students also described profound personal experiences of psychological distress (Table 5).

#### 3.2.2. Polyphasia in the Media

In the media, considerations of who, what and where were commonly tied in representation to imply the latent question: why are ‘mental health’ and ‘illness’ relevant to the public? This question is largely rhetorical and instead focuses on reproducing what is already treated as common-sense by both the author and the readership. Indeed, whilst articles were concerned with ‘educating’ the public in knowledge regarding mental ill-health, such as specific risk factors (Table 4, ranks 6–9), through the groups, places and topics referenced, mental health and illness were represented broadly to portend the public’s continued existence (Table 3, ranks 1–8). For example, articles privileged the voices of parents who spoke posthumously about deceased students (Table 3, ranks 1, 2 and 4), as well as professionals in mental health, education and criminal services (Table 3, ranks 1, 4, 5, 7 and 8).

Whom the public was represented to be and why mental health and illness are relevant to them were implied by the topics discussed: authors commonly focused on inquests into fatal harm, either through suicide or murder (Table 3, ranks 1–8). These harms were localised in the public imagination as simultaneously proximal (e.g., “yards from the front door” [Table 3, rank 5]) and distal (e.g., “around the country”, [Table 3, rank 1]). This gave the articles a confusing sense of space and time. Mental health and illness and illness were both represented as an urgent personal concern and a threat beyond the limits of ego-centric perception (Table 3).

#### 3.2.3. Polyphasia in the Focus Groups

In contrast to the focus in the media on societal institutions (i.e., universities, courts, hospitals [Table 3, ranks 1–8]), in the focus groups, concerns for intergroup contact focused on perceived interpersonal asymmetries (Table 4, ranks 1–5 and 7–9). For example, FG5P2 experienced “double standards” in cleaning to elicit interpersonal conflict and hatred (Table 4, rank 8). Similarly, FG4P3 represented mental health and illness as a deficiency in awareness to be remediated predominately through contact with mental health professionals (Table 4, rank 4). Concerns expressed in focus groups were localised to routinised actions, such as treatment adherence, cleaning, exercise and work (Table 4, ranks 1 and 6–9). This represented intergroup contact as a functional issue linking perception, action and effect (Table 4, ranks 1–10). This is illustrated by the description of mental illness as “appearing sad” and treatment as rendering “problems” imperceptible (Table 4, rank 10).

#### 3.2.4. Polyphasia in the Interviews

Representations developed in the interviews (Table 5) responded to societal and interpersonal concerns advanced in the media (Table 3) and focus groups (Table 4). In the media and focus groups, concerns for intergroup contact largely focused on the perceived deficiencies or vulnerabilities present in the out-group (Table 3 and Table 4). In the interviews, concerns for the out-group were also localised to concerns for the Self (Table 5). For example, P8 described their own “personal experiences” of feeling for months like “utter trash” to the point that it “starts to impact [their] daily life” (Table 5, rank 9). Similarly, in the one-to-one context of the interviews, P5 was oriented toward introspection and explained how they relive “past failures” (Table 5, rank 7). Yet, it is important to highlight how the commonality of experience did not necessarily translate into empathy. Namely, despite profound personal experiences of distress, there was an incredible ambivalence regarding the comprehensibility of mental illness. For example, P7 rhetorically asks: “how do you measure stress on the brain? There is no way to measure that” (Table 5, rank 6).

### 3.3. Group Identity

#### 3.3.1. Summary

We found motivated beliefs in group differences were infused into the multiple contextualised ways of representing mental health and illness. Indeed, Othering was pervasive across social contexts and epistemology. Othering functioned to preserve in-group representations of mental health and illness as ‘not-me’. Furthermore, although unified through a global practise of Othering, we found evidence in the data for unique relations between stigma and epistemology. Namely, people drew on a static epistemology to degenerate individuals with experiences of mental illness as a comprehensible and undesired social category, often drawing on professionalised language. Alternatively, people drew on a dynamic epistemology to represent experiences of mental health and illness as an incomprehensible threat realised in the uncertain relations between people and space.

#### 3.3.2. Othering

Across social contexts, mental health and illness were represented as Other. Often, expressions of Otherness were subtle or implied. P1 rendered this latent practise manifest (Table 5, rank 5). P1 differentiated and degenerated individuals with experiences of mental health and illness from themselves for their perceived lack of happiness and confidence (Table 5, rank 5). This perceived deficiency was represented as a permanent issue rooted in personhood: P1 represented mental ill-health as “deep-down” unhappiness in oneself (Table 5, rank 5). This clearly illustrates how Self-knowledge can be foundational for the representation of mental illness as Other.

By conceptualising mental health-related stigma as a Self/Other dialogue, we can critically evaluate P1′s claim that “if I was a friend, I wouldn’t feel annoyed” (Table 5, rank 5). P1′s argument is the foundation of dominant mental health-related anti-stigma campaigns, such as Time-to-Change. Although Self/Other dialogues may enable an ethic of mutual support, there is little to suggest that pro-social behaviours cross perceived in-out group lines. Our analysis of P1 highlights the conditional nature of lay ethical statements about mental health and illness (Table 5, rank 5). P1 represented individuals with experiences of mental illness as a group separable from themselves and their friendship or in-group (Table 5, rank 5). This means that whilst P1 may be supportive of a friend’s mental health-related challenges, this does not overcome their representation of mental illness as Other (Table 5, rank 5), undermining the basis on which contact-based anti-stigma is premised.

Reading P1 together with focus group six (Table 4, rank 9) and Buzzfeed (Table 3, rank 6) provides insight into the remarkable ambivalences that Other mental illnesses. In the media, experiences of mental illness were represented as risking societal exclusion. Buzzfeed’s headline represents the communication of “mental health” to risk one’s forced removal from university accommodation (Table 3, rank 6). Similarly, in focus group 6, individuals with experiences of mental illness were represented as constituting “another party” who would be incapable of normative behaviours such as “chores”, “cleaning” and paying “bills” (Table 4, rank 9). Furthermore, FG6P2 believed that anything could render manifest individuals with experiences of mental illness perceived underlying incapacities. For example, “an image, a smell, whatever it might be” may “bring all of the past experience to the fore” (Table 4, rank 9). As FG6P2 represented anything and everything to be a risk, and experiences of mental illness were largely assumed to be permanent, the possibility for friendship across group lines is muted (Table 4, rank 9). Instead, it was taken for granted that the relations between individuals with and without experiences of mental illness are asymmetric.

#### 3.3.3. Static Epistemology and the Other

People also Othered mental health and illness using a static epistemology. This degenerated individuals with experiences of mental illness to holding a reduced and deficient form of personhood. For example, in the media, students were disproportionately represented as ‘at-risk’ and in need of ‘protection’ (Table 3, ranks 3 and 6–9). Correspondingly, in the focus groups (Table 4) and interviews (Table 5), students emphasised the deficiencies in insight, awareness and capability between those perceived to have or not have a mental illness, as well as differences between more and less ‘extreme’ perceived disorder and treatment types. In each corpus, on account of their perceived ‘vulnerabilities’, individuals with experiences of mental illness were represented as dependent on ‘responsible’ groups, such as mental health professionals, family members and close friends (Table 3, Table 4 and Table 5). By emphasising the perceived deficiencies of individuals with experiences of mental illness, the public arguably sustained positive in-group representations as capable, agentic and aware.

People commonly expressed their static knowledge about mental health and illness using professional terminology (Table 3, Table 4 and Table 5). The presence of professionalised vernacular in lay thinking likely responded to social desirability as it obscured the immediate recognition of mental health-related stigma [92]. This is not to neglect how professionalised models may support individuals in distress. However, by defining mental illness in terms of deficiency, it limits our ability to challenge the power inequalities that reproduce and sustain stigma in society, such as those between providers and recipients of care [93]. Moreover, it conceals the diverse communications, such as humour, through which people live and grow during distress and prosperity [83].

## 4. Discussion

To review, rather than lacking ‘knowledge’ [12], the public made sense of mental health and illness using dynamic and static epistemologies [71]. There was little to suggest people experienced these alternative forms of knowledge as a source of tension [38,70]. Instead, people’s sense-making was characterised by polyphasia, such that representations of mental health and illness were functional in context [38,70]. Whilst people engaged in multiple contextually specific forms of knowledge, this did not overcome a group-based belief in mental illness as Other [74]. Furthermore, rather than considering mental health-related knowledge in the abstract, representations of the Other were a profound source of Self-knowledge [62,94,95].

The polyphasic nature of the public’s sense-making practises has critical implications for the continued utility of dominant mental health-related anti-stigma campaigns. As described, public health practitioners have predominately assumed stigma to be sustained by the public’s deficiencies in an abstract form of professional knowledge [2,12]. In contrast, through mental health-related epistemologies infused with stigma, people represented themselves and others in context [71]. From this perspective, although clearly degrading, the group-based identities proscribed by stigma held important functions in society, which may in part explain the difficulty of sustained mental health-related anti-stigma behavioural change [2,5,73].

Especially in the media, intergroup contact is represented as the risk of a loss of life [92,93,96]. Othering was functional in managing this perceived harm [50]. Despite profound personal experiences of psychological distress and the experience of social contact and professionalised knowledge, in the interviews and focus groups, students degenerated and differentiated themselves from individuals with experiences of mental illness [28]. This representational practise is typical of social categories represented as a *“matter of out of place”* (Douglas, 1966, p.44). To retain their perceived sanctity, in-groups commonly represent marginalised groups as ‘different’ and ‘separate’ [50].

Othering was engrained into the lay epistemologies used to make sense of mental health and illness [25,26,41]. Rather than constituting a ‘bastardised’ form of professional knowledge [65], dynamic and static explanations of mental illness were rich with social-historical meaning [27]. Dynamic accounts of stigma emphasise the perceived unpredictability and uncertainty associated with experiences of mental ill-health [2,26,74]. We argue that these meanings, and the dis-ease they elicit within Self, are rooted in culture and, in particular, continued public representations of the asylum [27,42,43,51]. Especially in Hollywood (e.g., One Flew Over the Cuckoo’s Nest, 1975), asylums are commonly caricatured as a place of horror, violence and constraint, which threaten the public [51,97]. In the British context, some of these former asylums have now been repurposed into mental health and educational services, including student accommodation [43]. Others became brownfield sites, some of which have been developed into ‘horror houses’ and luxury accommodation [43]. Many others remain in their abandoned state [43]. Indeed, ‘spirits’ and ‘demons’ are commonly represented as haunting those who ‘trespass’ onto the sites of former asylums [43]. Static accounts of stigma are likely also sustained through representations of the asylum. Coexisting with representations that sensationalise the asylum as a place of unpredictable violence, narrators have also characterised the asylum as a place of lethargy and routinised banality [41]. This likely enables the public to Other mental illness as something ‘fixed’ in the past [41].

Efforts to improve mental health-related stigma have focused on incremental improvements in content and delivery [10,11,15,34]. Some researchers have proposed that continuum models of mental health and illness may be more effective than categorical models, especially when delivered by someone with experience of a mental illness [15]. Overall, cross-sectional and experimental manipulations of continuum vs. non-continuum beliefs suggest that they have small mixed effects on public stigma [15]. Qualitative studies suggest that the ‘feeling of us’ may be central to explaining the variability in intervention effectiveness [15]. Unfortunately, our data questions how achievable interventions may be in encouraging this ‘feeling of us’. Namely, as representations were overwhelmingly negative and emphasised the perceived risks of harm to the Self, it is questionable how far an ethic of mutual support might realistically transcend perceived in-out group boundaries [94].

Whilst our findings confirm the importance of contact in making sense of mental health and illness, there is little to suggest that contact may be a sufficient ingredient for social and behavioural change [32]. Representations of contact were ambivalent and multiform [28,32,58]. We found people to be tenacious in distancing mental health problems beyond the boundaries of ego-centric perception [98], despite personal or parasocial contact with experiences of mental ill-health [4]. Moreover, whilst there were expressions of support for friends and family, this ethic was conditional upon perceived group membership [99]. In other words, whilst people might support a friend in distress, they also avoid developing friendships with someone they perceive to have a mental health problem [28,31]. This ambivalence may explain some of the mixed effects in the literature on social contact [32]. Whilst repeatedly nationally representative cross-sectional studies find prior contact to be correlated with lower degrees of stigma, a dose-relationship between contact-based interventions and stigma reduction is yet to be established [33] and evaluations of stand-alone contact-based interventions find mixed to no effects [14,22,100]. This might mean that whilst the language of contact is embedded in lay understandings of mental health and illness [26,58,83], contact may not be sufficient for social and behavioural change [16,33]. Indeed, what might appear as a material issue of contact, may actually conceal a latent issue of public motivation.

As mental health-related stigma constitutes a functional form of knowledge in society, we should question how productive it is to continue to demarcate mental health-related stigma as a narrow category of public health concern [17,73,93,101]. Instead, what might be of benefit is a greater distinction between education-based interventions and efforts to challenge stigma conducted in an educational setting [79]. Instead of conceptualising mental health-related stigma in terms of the public’s deficiencies in abstract knowledge, we might draw on ecological theories and ask how social contexts enable or challenge a group’s representations of mental illness as Other [16,28,53,95]. From this perspective, educational centres might be conceptualised as major sites for socialisation [67,102], in which stigma is reproduced in the interpretive engagements of social actors, including social meanings, affective states, roles and ideal types [5].

Moves have already been made in this direction, especially through the emergence of whole university approaches to stigma [103,104]. Rather than mental health support being a stand-alone service provided by a specialist team, whole university approaches to mental health and illness integrate all aspects of university life–from the design of curricula and assessments to the built environment [103,104].

When taking an ecological approach to stigma, mutual support groups may constitute an underutilised means for lay stigma alleviation [8,84,105]. Although traditionally preferred by service users, as our studies highlighted the centrality of Self-knowledge in sustaining mental health related-stigma, mutual support groups may provide a space for students to collectivise experiences of distress and advocate for alternative understandings of mental health and illness [8,95].

However, we caution against providing simplified statements on how to ‘correct’ anti-stigma efforts, as we acknowledge that even in service user mutual support groups, mental illness is regularly Othered [95]. Indeed, whilst we hope that our account of theories of cognitive polyphasia, social context and group identity will enable practitioners to engage with mental health-related stigma, given the sheer complexity and sensitivity of the ways the public communicates mental health and illness, we do not proscribe generalised interventions [58]. Ultimately, effective social and behavioural change programmes are likely developed within and remain unique to the particular group and social context of concern [58,60,106].

## 5. Conclusions

Education-based interventions conducted with or without- social contact constitute the dominant public health approach to mental health-related anti-stigma [2]. Interventions aim to remediate the public’s assumed deficiencies in abstract professionalised knowledge [2,12]. However, there is limited evidence that interventions cause sustained social and behavioural changes, and there is a serious concern for their likely unintended consequences [2,8,14]. Our research revealed the flawed basis on which campaigns are conceived [12]. Rather than a deficiency in knowledge, we found people engaged in multiple dynamic and static epistemologies, which were functional in the social context [26,38,70]. Moreover, knowledge about mental health and illness was motivated [68]: to protect the Self against the perceived risks posed by contact with mental ill-health, irrespective of social context or epistemology, people Othered individuals with experiences of mental illness [28,50]. This has profound implications for the continued utility of current education and contact-based anti-stigma strategies and highlights the need to explore alternative means for stigma alleviation [2]. It is our hope that by engaging with ecological theories we can develop improved strategies responsive to the public’s multiple common-sense understandings of mental health and illness [58].

## Figures and Tables

**Table 1 ijerph-19-10618-t001:** Newspaper Frequency Statistics.

Media Outlet	Frequency
The Guardian/Observer	119
Google News	93
Google Search	92
Daily Mail	54
LADBible	41
BBC	37
Youtube News	24
ITV	23
Buzzfeed	19
Sky News	18
Channel 4 News	9

**Table 2 ijerph-19-10618-t002:** Participant demographics.

		Interviews (N = 18)	Focus Groups (N = 20)
**Gender**	Male	7	3
	Female	11	17
**Age**	Mean	22	23
	Range	18–30	18–35
**Qualification**	Undergraduate	13	10
	Postgraduate	5	10
**Subject**	Psychological and Health Sciences	7	10
	Social Sciences and Humanities	6	7
	Natural Sciences	5	3
**Citizenship**	UK	11	14
	EU	7	6

**Table 3 ijerph-19-10618-t003:** Media TF-IDF rank score quote table.

TF-IDF Rank	Word	Verbatim Quote
1	Students	“Natasha’s parents have had to turn to the civil courts to seek justice for their daughter. They are determined to try to improve the standard of care provided to vulnerable students around the country.” (The Guardian, 19 July 2020)
2	University	“The university is overselling a gilded version of student life to woo teenagers—some of whom are not mature enough to cope—in order to land annual tuition fees of £9250. The mother of one victim complained … that Bristol is trying to ‘cover up’ the deaths.” (The Sun via Google News, 18 June 2018)
3	Mental	“**Universities need to do more to protect students’ mental health. But how?** [emphasis original] They are less happy than the general population, depression and loneliness now affect one in three of them, and the number of suicides among this group is rising.” (The Guardian, 22 July 2016)
4	Failure	“In a statement read at the beginning of the inquest, Natasha’s mother said she thought that a previously high achieving student would have seen this as “a huge failure” (The Mirror via Google News, 16 May 2019)
5	Year	“The defendant told psychiatrists that he started to receive telepathic messages and considered himself the ‘chosen one’ or ‘Messiah’ in spring last year… **Dr Ensink was killed just yards from his front door shortly after Christmas last year.** [emphasis original] Nandcap had being caught in May last year and allegedly in possession of two kitchen knives and assaulting a police officer, who he punched and bit on the thumb.” (The Daily Mail, 11 October 2016)
6	Risk	“**Suicidal student not to discuss his mental health with others or risk being kicked out of student residence**” [Emphasis original] (Buzzfeed, 17 May 2016)
7	Suicide	“At the same time we know that Natasha was being badly let down by specialist mental health services who failed to put in place a timely and adequate plan to mitigate Natasha’s risk of suicide.” (The Daily Mail, 16 May 2019)
8	Depression	“Mental health professionals have also warned that the sight of the tower from the busy Westway Road also risks traumatising a far wider population, in particular children… thousands could suffer post traumatic stress disorder, which can cause panic, anxiety, depression and flashbacks.” (Guardian, 14 December 2017)
9	Help	“With students facing rising mental health concerns and pressures, digital data analytics systems could help universities identify those at risk and provide a timely response” (The Guardian, 8 March 2019)
10	Housing	“Housing stress is a major cause of mental health issues, yet for the students at UCL it is their own university, a centre of education, that is imposing this upon them… We have the potential to change the balance of the current dynamic in the housing market back to favour the tenants and families currently bearing the brunt of the housing disaster.” (The Guardian, 27 January 2016)

**Table 4 ijerph-19-10618-t004:** Focus group TF-IDF rank score quote table.

TF-IDF Rank	Word	Verbatim Quote
1	Impact	“FG6P2: if you’ve got this person in your environment at work it’s like, “Okay, you’re going to have to work harder because they can’t work,” or you know, they’re having a day off. I think people could be resentful of this very, very quickly. In the same way, if you’re like having your living-in uh, situation impacted by someone on a relatively regular basis, I think people could feel that way.” (Focus Group 6)
2	Treatment	“FG2P2: They [professionals] can also give some instructions for the patients to, um, I know, well, steps to treat, well, a treatment--for a person in order to--for-for the problem to disappear.” (Focus Group 2)
3	Environment	“FG3P2: I think it’s really hard to stop that because if you meet someone you don’t know and he can say something--or you might interpret it in a different way. Yeah, there’s a lot of environmental triggers like you know, you--media and all that stuff.” (Focus Group 3)
4	Awareness	“FG4P3: I think when it comes to awareness is really subjective because it can really--I don’t know it just very--It is quite flexible because some people generally don’t know at all and may struggle to know until someone like a professional or even close family friend or friend in general will tell them what’s actually going on.” (Focus Group 4)
5	Therapy	“FG5P1: So the first line is, um, typically that we follow the NICE Guidelines… And typically the first line of treatment is psychotherapy- And then if like nothing is sort of like happening, and then maybe try medication… I think there’s a like a harder, uh, connotation with medication-than therapy. Therapy feels soft, doesn’t it. -as opposed to drugs.” (Focus Group 5)
6	Anxious	“FG3P3: You’re very stressed that day or very anxious that day and it makes you feel better to do some exercise which you can offload from something… fluctuations in mental health can be problematic if you’re like oscillating between very different extremes of feelings.” (Focus Group 3)
7	Drugs	“FG6P3: I would not know what to do. FG6P1: you can also genuinely help people with like depression or anxiety, whereas with schizophrenia, I think the only real solution is drugs, right?” (Focus Group 6)
8	Extreme	“FG5P2: They’re already a mess, but they expect you to be extremely clean… It’s why I really--Don’t really like her, I hate her. Yeah. There’s double standards involved.” (Focus Group 5)
9	Brings	“FG6P2: something in their past or experience that is exacerbated or renewed by a behavior, a presence an image, a smell, whatever it might be there is. It brings all of the past experience to the fore. [chuckles] FG6P1: People may feel like the other party is gonna not share some of the chores, like take out the garbage, for example. And then maybe they’ll be too depressed to clean, take out the trash, do things like that. FG6P2: Or pay the bills.” (Focus Group 6)
10	Loss	“FG1P1: like lack of motivation, like, just like, appearing sad, you know? That kind of, like, loss of something.” (Focus Group 1)

**Table 5 ijerph-19-10618-t005:** Interview TF-IDF Rank Score Quote Table.

TF-IDF Rank	Word	Verbatim Quote
1	Disorder	P4: “their first image of a mental disorder would be depression, suicidal tendencies and stuff like that.” (Interview 4)
2	Behaviour	P18: “it’s the way things are expressed in terms of behaviour and voice and action. It’s not just thinking, it’s how things are expressed. It’s quite difficult. I’ve known other people who see things so innocently. They are so innocent and almost naive, but that’s just how they rationalize things.” (Interview 18)
3	Children	P3: “With severe ones, they’re putting the risk on everybody. You don’t know what they can do. They could potentially harm you or your children, or it could be a risk to the family… you do not want to discuss anything that’s very sensitive… They could overreact, they don’t know boundaries, they could do anything, like so these are things to associate with mental illness.” (Interview 3)
4	Understand	P14: “You don’t understand why they’re thinking that… You might start thinking negatively about other things as well if they just have that negative perspective.” (Interview 14)
5	Difference	P1: “If I was a friend, I wouldn’t feel annoyed… It’s hard to explain. To them I think when they’re all closed up, they want to be closed up but deep down they’re just unhappy with themselves that’s why they’re closing themselves up. For me, I’m happy in myself, I’m confident.” (Interview 1)
6	Stress	P7: “how do you measure the stress on the brain? There is no way to measure that.” (Interview 7)
7	Mood	P5: “I even have random mood swings sometimes like I’ll be happy for one second and then literally everything gets cloudy and stuff. I feel like--everything feels hopeless. You give up hope in yourself in those moments… and I just think of past failures, and I’m like, what’s the point of doing this” (Interview 5)
8	Social	P6: “Unclean. Very anti-social. Also being like inconsiderate of other people especially at night so having friends over and things like that” (Interview 6)
9	Difficult	P8: “if those were the two poles then the middle bit is like very difficult to distinguish like. I know from personal experience, I’ve gone through periods where, weeks at time capping in up to after two months, I can feel like utter trash and it starts to impact my daily life.” (Interview 9)
10	Awareness	P10: “It’s also a lot of practice on self-awareness and things like that. Just like how someone who is normal, who fell sick, apart from taking medication, in the long run, in order for them not to fall back into getting fever” (Interview 10)

## Data Availability

The data are not publicly available to maintain participant confidentiality, as this paper concerns sensitive personal information. This was agreed with university ethics committee, and it was set out in informed consent procedures. Anonymised verbatim quotations were used to maintain research transparency.

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
