# Peer review of "Charting an Alternative Course for Mental Health-Related Anti-Stigma Social and Behaviour Change Programmes"

_ijerph, 2022, doi:10.3390/ijerph191710618_

Round 1

Reviewer 1 Report

The manuscript “Charting an alternative course for mental health-related anti- 2 stigma social and behaviour change programmes" presents an interesting approach to the collection information of the Mental health.

The “Introduction” section of the manuscript provide extensive revision and with a very good redaction.  The review of literature  is relevant to their study.

The aim of the study is properly highlighted and justified.

The manuscript, using mixed techniques based in cuestionnaires and focus group. The presentation of the technique and characterization of the results achieved indicate that the method is quite suitable and in fact could be useful to profundize this aspects in the mental health. However, I would like to receive more information about the data collection:

1. were the participants paid?

2. did they sign the informed consent?

3. did the project pass through an ethics committee?

4. Was it a convenience sample or what type of sample?

Overall the results are compelling and indicate that the method is more than suitable. Rigorously,  the analisys are  detailed.  The authors do a very good job of presenting a methodology of the accuracy and precision of their results and demonstrate the suitability of the method. Further, the manuscript presents a good and actualized bibliography.  The study is of interest for the scientific community.

Reviewer 2 Report

Comments:

TF-IDF stands for Term-Frequency Inverse-Document Frequency, not Inverse-Term Frequency.

Line 129: delete "an"

Section 2.1.1 -- the included news reports do not constitute a "sample" in the statistical sense, since the all news reports satisfying certain criteria were all included. I suggest replacing "sampled" with "included", or some such word that does not suggest a statistically valid sample. 

Line 252: How were the 38 students recruited? On what basis was 38 chosen as an adequate number to recruit?

Line 322: What software was used to apply the TF-IDF algorithm? 

Reviewer 3 Report

The authors have well described the need for alternative strategies for decreasing the stigmas associated with mental health and as such provided an evaluation through news reports, focus groups and interviews.   Through their analyses, they argue for the development of anti-stigma strategies based on sense-making strategies and ecological approaches to address the stigma of mental health.

Overall, I feel that this manuscript is well-written and worthy of publication.  

Lines 92 to 93: Please confirm the most recent acronym is appropriate and inclusive (e.g. LGBT versus LGBTQIA+).
